# Theta oscillations locked to intended actions rhythmically modulate perception

Alice Tomassini*, Luca Ambrogioni, W Pieter Medendorp, Eric Maris*

Donders Institute for Brain, Cognition and Behavior, Centre for Cognition, Radboud University, Nijmegen, Netherlands

**Abstract** Ongoing brain oscillations are known to influence perception, and to be *reset* by exogenous stimulations. Voluntary action is also accompanied by prominent rhythmic activity, and recent behavioral evidence suggests that this might be coupled with perception. Here, we reveal the neurophysiological underpinnings of this sensorimotor coupling in humans. We link the trial-by-trial dynamics of EEG oscillatory activity during movement preparation to the corresponding dynamics in perception, for two *unrelated* visual and motor tasks. The phase of theta oscillations (~4 Hz) predicts perceptual performance, even >1 s before movement. Moreover, theta oscillations are phase-locked to the onset of the movement. Remarkably, the alignment of theta phase and its perceptual relevance unfold with similar non-monotonic profiles, suggesting their relatedness. The present work shows that perception and movement initiation are automatically synchronized since the early stages of motor planning through neuronal oscillatory activity in the theta range.

## Introduction

Our motor system orchestrates the sampling of sensory information by orienting our receptor organs in space and time. Mounting evidence further suggests that motor signals, such as corollary discharges, also contribute to the actual analysis of the incoming sensory data, thereby shaping perception (e.g., [*Schroeder et al., 2010*; *Morrone et al., 2005*; *Rolfs et al., 2013*; *Tomassini and Morrone, 2016*; *Schubotz, 2007*; *Tomassini et al., 2014*]).

We recently provided behavioral evidence that low-level visual function is rhythmically modulated during preparation for a voluntary movement of the arm (*Tomassini et al., 2015*). In fact, we found theta-band fluctuations of visual contrast sensitivity that are time-locked to the movement and emerge before its onset. Crucially, this modulation affects visual stimuli that are *unrelated* to the motor task (*Tomassini et al., 2015*; *Benedetto et al., 2016*), extending traditional notions of the corollary discharge (*Crapse and Sommer, 2008*). While these results suggest a rather automatic form of sensorimotor coupling, the neural mechanism underlying this coupling has remained unclear.

A recent series of studies has shown that the phase of ongoing brain oscillations just prior to stimulus presentation influences its subsequent perception (*Busch et al., 2009*; *Mathewson et al., 2009*; *Hanslmayr et al., 2013*; *Busch and VanRullen, 2010*). Furthermore, the ongoing rhythmic activity is flexible, can be under attentional control (*Landau et al., 2015*; *Cravo et al., 2013*; *Samaha et al., 2015*; *Rohenkohl and Nobre, 2011*), and is susceptible to a phase-reset by external events - even of a different modality (*Romei et al., 2012*; *Mercier et al., 2013*; *Lakatos et al., 2009*). An intriguing possibility is that action planning is accompanied by an endogenous phase modulation of perceptually-relevant rhythmic activity, resulting in the synchronization of perception and movement initiation. This oscillation-based synchronization would explain the movement-locked fluctuations in visual sensitivity described above (*Tomassini et al., 2015*; *Benedetto et al., 2016*). The present study was set up to address this specific hypothesis by investigating the trial-by-trial

*For correspondence:
a.tomassini@donders.ru.nl (AT);
e.maris@donders.ru.nl (EM)

**Competing interests:** The authors declare that no competing interests exist.

dynamics of oscillatory activity during motor preparation in relation to the corresponding dynamics in perceptual performance.

## Results

In two experiments, one behavioral and one combined behavioral-EEG, we collected data from six and seventeen participants, respectively, performing the same dual task.

Participants fixated on a central cross that briefly changed color to indicate one of two waiting periods (1.5 or 2.3 s) after which they had to push an isometric joystick with their right hand (*Figure 1*; see also Materials and methods). At random times around the instructed movement time (from – 0.35 to +0.25 s) a near-threshold Gabor patch, tilted by ±45 deg, was briefly displayed at the center of the screen and participants had to report its orientation (clockwise or counterclockwise) verbally (as in [*Tomassini et al., 2015*]).

To investigate the coupling between motor and perceptual processes, we aligned the visual performance (% correct) to movement onset. This revealed large rhythmic fluctuations, beginning during motor preparation and continuing during motor execution (*Figure 2a*), which confirmed previous findings (*Tomassini et al., 2015*; *Benedetto et al., 2016*). We identified the spectral content of this movement-locked rhythmicity in perceptual performance with a random-effects analysis based on logistic regression (see *Figure 2c* and Materials and methods). This analysis consists in testing whether a sinusoidal function with the same frequency (in a range from 2.5 to 14.5 Hz) and phase across participants significantly predicts the perceptual outcome.

A prominent theta-band component, centered at ~4 Hz, is present in the perceptual time courses of both the purely behavioral and the combined behavioral-EEG dataset (*Figure 2b*). A second component, in the alpha-band and centered at 10 Hz, is evident only for the purely behavioral dataset. This component may be obscured in the EEG dataset due to the smaller number of trials and the consequent sparser sampling along the time axis. In our analysis of the EEG data, we therefore focus on the ~4 Hz component, which is also in line with the previous reports of movement-locked perceptual periodicity in the theta-band (*Tomassini et al., 2015*; *Benedetto et al., 2016*).

The rhythmicity in perception aligned to movement onset suggests the existence of a corresponding brain rhythm whose phase is predictive of the perceptual outcome. To test this hypothesis, we estimated the instantaneous theta phases in a ~2 s window prior to movement onset and statistically evaluated their predictive power for perception as a function of time relative to movement onset. To do so, on a trial-by-trial basis, we first extrapolated to stimulus onset the phases calculated during motor preparation (see *Figure 3c* and Materials and methods). The latter (extrapolated) phases were used as predictors of the perceptual performance, in the same way as we did for the analysis of the behavioral time courses. We evaluated the association between the measured oscillatory phase and the perceptual outcome, and statistically tested its consistency across participants (i.e., same highest/lowest-performance phase; see Materials and methods).

As hypothesized, the theta phase time-locked to movement onset reliably predicts perceptual performance (*Figure 3a*). Strikingly, the time course of the predictive value of the theta phase during motor preparation shows a peculiar, non-monotonic profile: an early peak with statistically significant predictive values more than 1 s before movement (~−1.4 s) is followed by a non-predictive period and a later monotonic increase as the time of movement approaches. Importantly, most of the stimuli were presented close to movement onset, between −0.5 and +0.5 s (see *Figure 3c* for the distribution of stimulus presentation times). Thus, contrary to the late effect, the early predictive peak reflects a remote influence, whereby the phase of theta oscillations is informative of the perceptual fate of stimuli occurring more than 1 s later.

To gather further insight on the temporal dynamics of theta oscillatory activity during motor preparation, we next looked at whether theta oscillations are phase-locked to the future intended movement, as predicted by our hypothesis. We thus quantified the inter-trial phase-locking in the pre-movement epoch and evaluated it statistically by means of a random-effect analysis based on a measure of phase reliability at the single-subject level (see *Figure 3f* and Materials and methods for explanation).

*Figure 3d* shows that theta oscillations are indeed reliably phase-locked to movement onset. Crucially, the time course of the theta phase-locking (panel d) closely matches that of the predictive value of the same oscillatory phase for perception (panel a). In fact, theta oscillations are aligned to

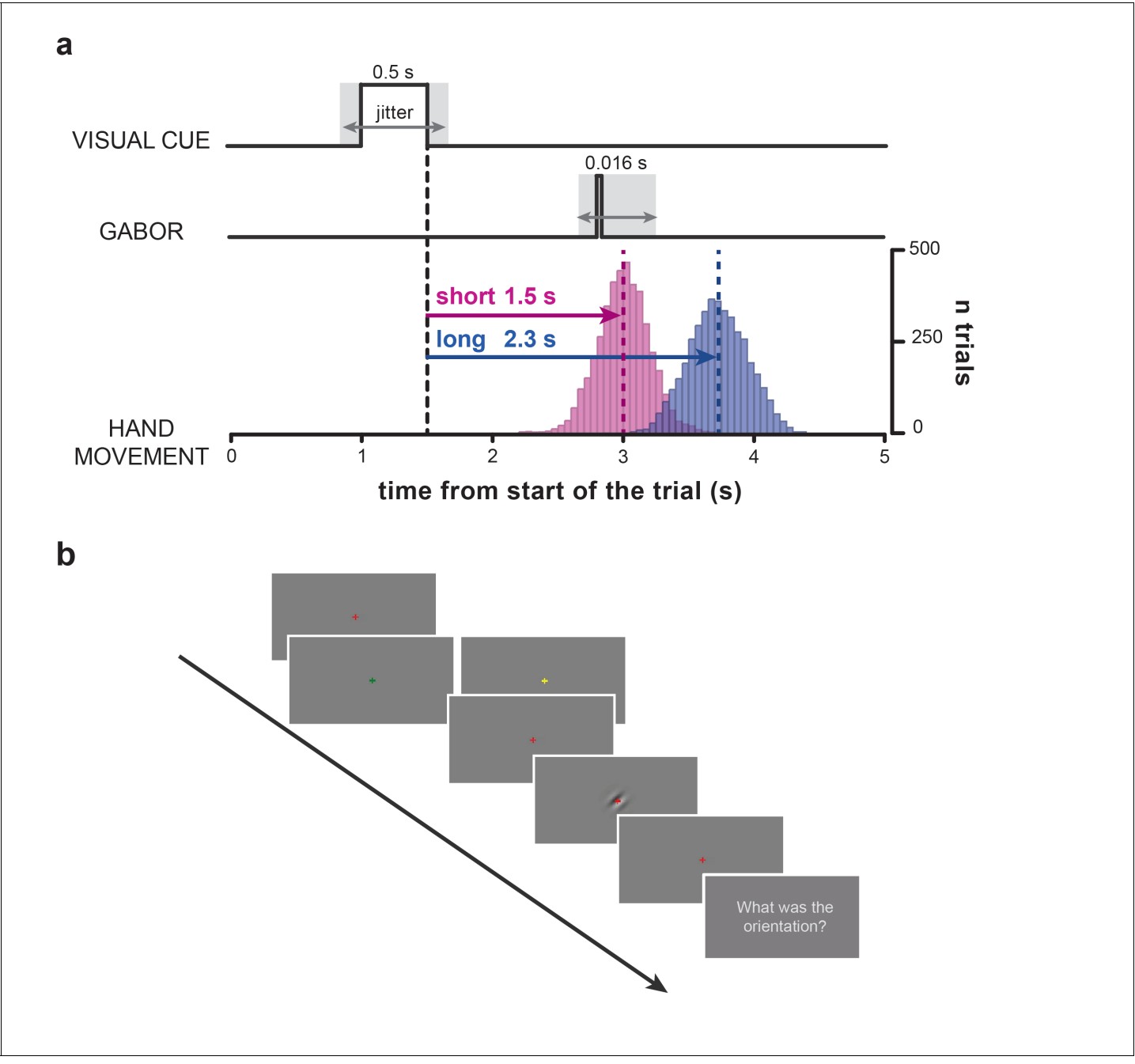

**Figure 1.** Task. (a) Timeline of the trial. After a variable delay between 0.8 and 1.2 s from the start of the trial (i.e., display of the red fixation cross) the visual cue is presented (i.e., a change in color of the fixation cross to either yellow or green). The color of the visual cue (counter-balanced across subjects) indicates whether the participants have to wait for a *short* (1.5 s) or a *long* (2.3 s) time interval before executing the hand movement with the isometric joystick. After 0.5 s, the visual cue is removed (i.e., the fixation cross turns red again). The offset of the visual cue marks the start of the time interval (black vertical dashed line) that participants have to wait before executing the hand movement. Bar histograms show the distribution of movement onset times (pooled across participants) for the short (pink) and long (blue) movement timing condition. The dashed vertical lines indicate the mean onset times (short: 1.5 ± 0.2 s; long: 2.22 ± 0.24 s; MEAN±SD). At random times between –0.35 and +0.25 s relative to the instructed movement time (*short* time interval in this example) a near-threshold contrast Gabor tilted 45 deg clockwise or counterclockwise is briefly flashed for 0.016 s (two frames). Therefore, cue-Gabor delays are on average 1.45 and 2.25 s for the short and long condition, respectively. (b) Example series of snapshots of the visual display during the trial. The red fixation cross is displayed throughout the trial over a uniform gray background, except when it changes color to either yellow or green (visual cue) to indicate the movement timing condition (short/long). The fourth snapshot shows a clockwise-tilted Gabor as an example (for illustrative purposes visual contrast is higher than what used in the experiment). Participants were instructed to wait for the appearance of the question 'What was the orientation?' before verbally reporting the orientation of the visual Gabor (last snapshot).

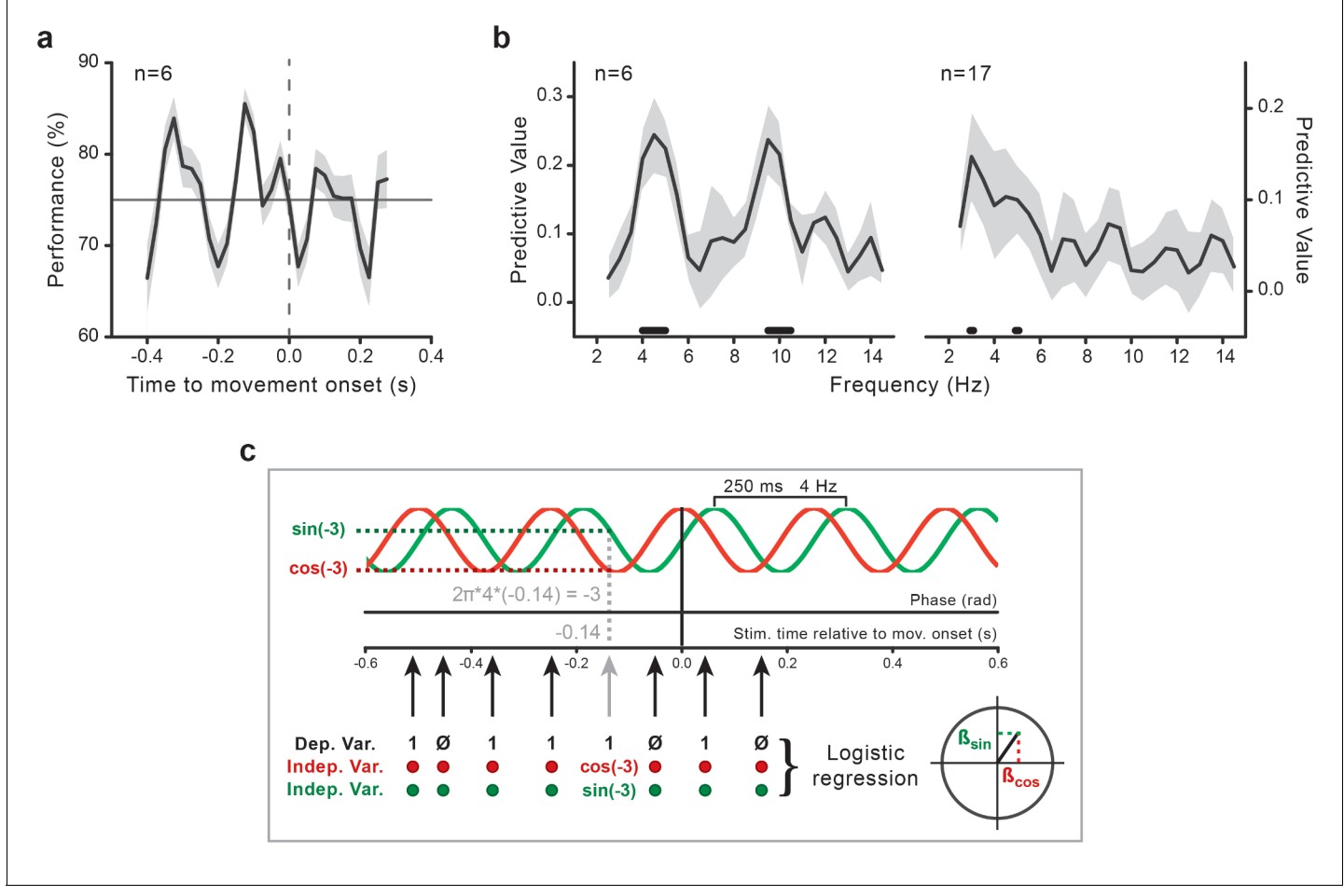

**Figure 2.** Movement-locked oscillations in visual perception. (a) Time course of the performance (percentage of correct responses) in the orientation task as a function of stimulus presentation time relative to movement onset (zero time by definition) calculated for the data pooled across subjects (n = 6, behavioral experiment). The gray-shaded area indicates the standard error. The horizontal line indicates threshold performance level (75% correct); the dashed vertical line indicates movement onset time. (b) Predictive value (estimated with Jackknife) for the perceptual outcome of sinusoidal functions with frequencies between 2.5 and 14.5 Hz for the purely behavioral (n = 6; left) and the combined behavioral-EEG dataset (n = 17; right). The gray-shaded area represents the jackknife standard error. The black horizontal bars indicate the frequencies that significantly predict perceptual performance at the group level (p<0.05, not corrected for multiple comparisons across frequencies; see Methods and panel c). (c) Schematic illustration of the analysis for an example sinusoidal function of 4 Hz. For each trial, the phase of the sinusoidal function at stimulus presentation time is computed as 2π*f*t (where f is the frequency of the sinusoid and t is the stimulus presentation time relative to movement onset; in this example 4 Hz and –0.14 s, respectively). The sine and the cosine of the resulting phase value (here, –3 rad) are then used as regressors (independent variables) to predict the perceptual performance (0–1, incorrect-correct) in a logistic regression analysis. Separate regression models are fitted for each subject and frequency in the range from 2.5 to 14.5 Hz. Second-level random-effect analysis is performed by submitting the participant-specific beta coefficients to the Hotelling's T-square test against zero (see Methods). This test provides significant results only if two conditions are concomitantly fulfilled: (1) the regression coefficients are large (i.e., the phase of the sinusoidal function is consistently associated with perceptual performance), and (2) they have the same sign across subjects (i.e., the phases associated with the highest/lowest performance are aligned across subjects). A very similar pattern of results is also found with a fixed-effects analysis approach based on fast Fourier transform (FFT) on the aggregated data from all participants in combination with permutations at the single trial level (see *Figure 2—figure supplement 1* and Materials and methods).

The following figure supplement is available for figure 2:

**Figure supplement 1.** Behavioral results obtained with a fixed-effect analysis based on fast Fourier transform (FFT).

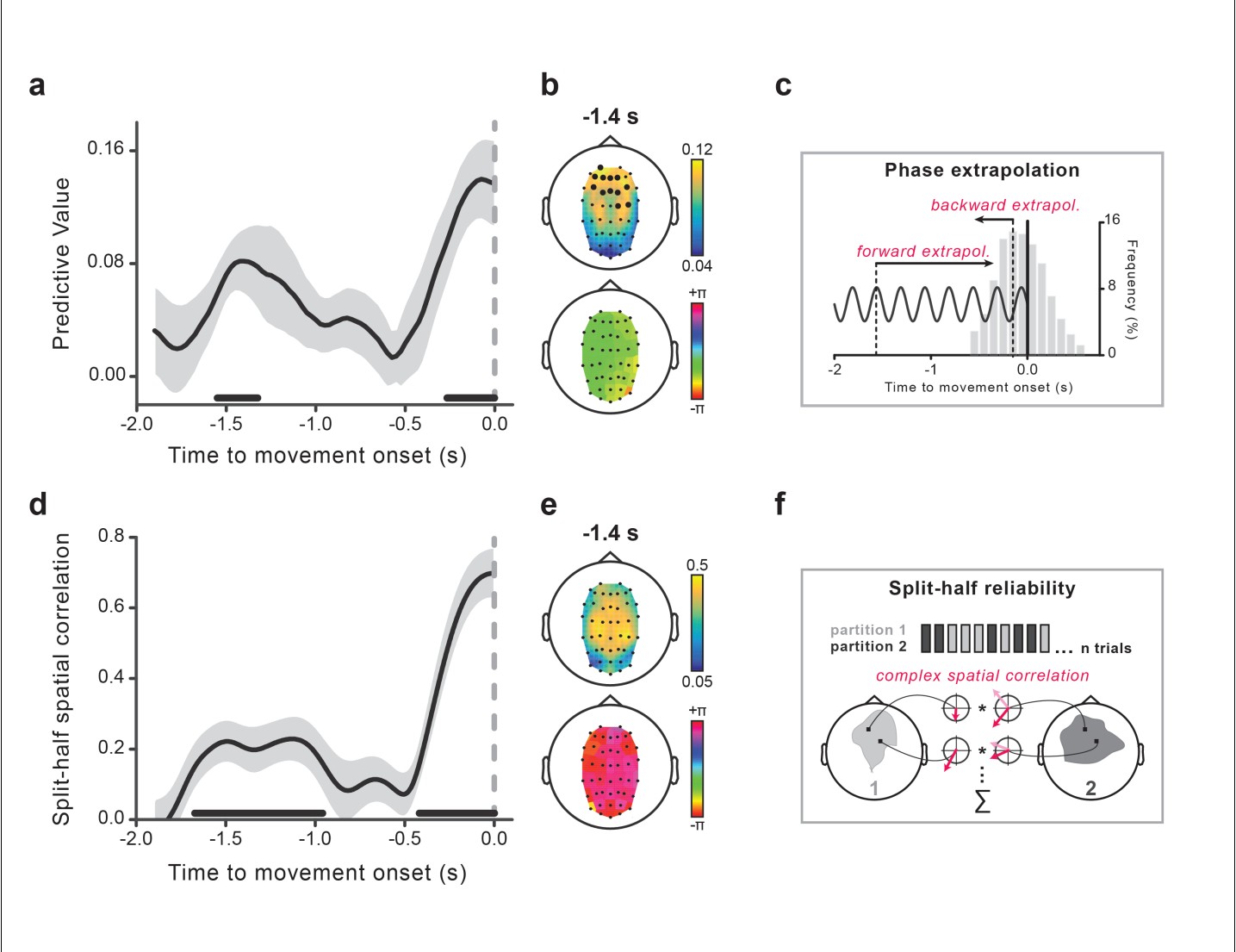

**Figure 3.** Theta phase predicts visual performance and aligns to movement onset. (a) Predictive value (estimated with Jackknife) of the 4 Hz theta phase for perception as a function of the time where the phase was estimated relative to movement onset. The gray-shaded area represents jackknife standard error. The black horizontal bars indicate significant time points (after False Discovery Rate – FDR correction across space and time). (b) The topography of the predictive value of theta phase for perception is shown at −1.4 s (earliest peak time in the effect) in the top graph. Significant channels are marked by bigger black circles (FDR-corrected). The topographical distribution of the optimal phase angle (the phase associated with the highest perceptual performance) is shown for the same time point (−1.4 s) in the bottom graph. (c) Schematic illustration of the phase extrapolation procedure. For each trial, the phase at stimulus onset is calculated by extrapolation from the instantaneous EEG phases estimated in the interval from −1.9 to 0 s relative to movement onset. For time points long before the movement (from −1.9 to ~−0.6 s) extrapolation is only performed forward in time, as the visual stimuli always follow the EEG phase estimation point. For points closer to movement onset, extrapolation can either be forward or backward in time, depending on the stimulus presentation time relative to the phase estimation point. The distribution of stimulus presentation times for all analyzed trials (between −0.6 and +0.6 s) is plotted in the gray bar histogram (pooled across subjects). (d) Time course of the split-half spatial correlation (see f) for the theta phase-locking to movement onset time. The gray-shaded area represents the standard error of the mean. The black horizontal bars indicate the time points where the theta oscillation is significantly phase-locked to movement onset (FDR-corrected). (e) Topography of the strength of the inter-individual consistency in the phase of the theta movement-locked oscillation (top) and topographical distribution of the mean phase angle (bottom) are shown for the same time point as in panel b. (f) Schematic illustration of the statistical assessment of the phase-locking to movement onset. For each subject, trials are split into two random partitions of equal size. The mean across trials of the signal's Hilbert transform time-locked to movement onset is calculated for each partition (and channel) - represented by the red vectors (*mean resultant vectors*, MRVs). The results are then correlated across space (see Materials and methods for details). This procedure is repeated 500 times, and the obtained spatial correlations are averaged across iterations. This single-subject correlation serves as the input for the group-level statistical test, which consists in submitting the real parts of the individual correlation values to a one-sample t-test (in fact, under the null hypothesis of no phase-locking to movement onset, the expected value of the real part of the complex-valued spatial correlation is equal to 0; see Materials and methods).

*Figure 3 continued on next page*

*Figure 3 continued*

The following figure supplements are available for figure 3:

**Figure supplement 1.** Early and late theta phases have independent predictive power for perception.

**Figure supplement 2.** Theta predictive value for perception is modulated by movement timing.

the future movement very early in time (~−1.4 s), phase-alignment then temporarily disappears, and is finally restored before movement initiation. Moreover, theta oscillations not only are aligned to the movement but they do so with a similar phase angle across participants. *Figure 3b–e* shows the topographies of the early effects (calculated at −1.4 s): theta phases that are most predictive for perception are concentrated over frontal electrodes (panel b, top), and theta phases that are most consistently locked to movement onset are concentrated over more central electrodes (panel e, top). Both the optimal phase angles (i.e., those associated with the highest perceptual performance) and the mean phase angles (expressing the phase relation with movement onset) are nearly identical across the channels (panel b-e, bottom).

The temporal discontinuity in the theta phases' predictive value and their alignment to movement onset raises a fundamental question about the nature of the underlying oscillatory phenomenon. Two scenarios are in principle possible. First, the discontinuity may result from a continuous oscillatory phenomenon whose sensor-level visibility is temporarily masked, possibly involving an intervening process. In this scenario, the early and the late phases would be coupled (coherent). Alternatively, the discontinuity may reflect two distinct processes, and in this scenario, the early and the late phases would be uncoupled (incoherent).

A straightforward way to distinguish between these two alternatives, is to determine whether the early and the late theta phases independently predict perception. To investigate this, we ran the same logistic regression analysis as used before, but now using as predictors both the early (estimated at −1.4 s) and the late (estimated at −0.1 s) theta phases, allowing their shared explanatory contribution to be discounted. Both the early and the late theta phases retained their original predictive value for perception, with no substantial change in effect size and topography (see *Figure 3— figure supplement 1*). This is clear evidence for two de-coupled oscillatory phenomena in the theta range.

As an additional piece of evidence, the early and the late effect also have different topographies, with concentration over, respectively, fronto-central and occipito-parietal sites (*Figure 3—figure supplement 1*). Although this topographic evidence is not conclusive by itself (because of rotating sources and travelling-wave-like phenomena), together with the results of the logistic regression analyses, it argues for a distinct origin of the early and the late effect.

Because of the motor timing component of our dual task, the onset of the movement follows the visual cue by a certain amount of time, which, for the *short* condition, almost coincides with the period at which the early theta effect is observed (cue-movement interval: 1.5 ± 0.2 s; MEAN±SD). To exclude that the early effect is actually induced by the visual cue (rather than being movement-related) and is uniquely present in the short trials, we ran separate analyses for the two timing conditions. Both sets of trials display a similar non-monotonic pattern in the predictive value of the theta phase (*Figure 3—figure supplement 2*). Interestingly, the early peak in the effect is higher, sharper and earlier for the long as compared to the short trials. A group-level permutation-based test (with FDR correction across space and time) confirmed that the temporal dynamics of the theta predictive value is consistently modulated by the motor timing condition. Specifically, this statistical analysis revealed a significant difference between the long and short trials at three different moments in time (−1.475 s, −1/–0.95 s and −0.6 s; see *Figure 3—figure supplement 2*). This pattern of results clearly rules out any potential confound of the visual cue: (1) the difference in peak latency between the short and long condition (~0.4 s) does not match the difference between the short and the long-time interval (0.8 s), as would be expected if the effect was caused by the visual cue, and (2) the peaks in the predictive value are broader for the short as compared to the long condition, whereas the reverse is expected if the effect is caused by the cue as a result of the larger variability in movement

onset times for the long as compared to the short trials ($t_{16}$ = −4.729, p<0.0001). Thus, the modulation of the predictive effect by the target movement time (long versus short) suggests that the perceptually relevant theta phases are generated by a process that is involved in movement timing.

Contrary to the early predictive effect, the late effect is observed at the same time as two other salient events: motor execution and Gabor presentation. Whereas the movement-related potentials could contaminate theta phase estimation, stimulus (Gabor)-evoked responses reflecting detection (e.g., the P300) could confer detection-related predictive power to the movement-locked theta phases. We therefore ran an additional analysis to rule out the potential confound by detection-related responses for the late predictive effect. Specifically, we considered only those trials in which the phase extrapolation is performed in the forward (not backward) direction, thereby analyzing only EEG signals preceding (not following) stimulus onset (see *Figure 3c*). These trials cannot be affected by stimulus-evoked responses. *Figure 4a–b* (top) shows that, although slightly weaker (but nevertheless statistically significant at −0.1 s, the time point of the maximum original effect), the predictive value calculated using only the (unconfounded) pre-stimulus phases closely resembles the one calculated for all trials; it does so with respect to temporal profile, topography and optimal phase angle. We therefore conclude that the post-stimulus phases do not contribute significantly to the late effect, which can hence be considered a genuine *predictive* effect (like the early effect).

It is interesting to know how the predictive effect of the movement-locked phases relates to that previously reported for the stimulus-locked phases (*Busch et al., 2009*; *Mathewson et al., 2009*; *Busch and VanRullen, 2010*). To investigate this, we calculated the predictive value of the theta phase as a function of stimulus (Gabor) onset time. Stimulus-locked phases (for all as well as for only pre-movement-stimuli, not corrupted by post-movement activity) significantly predict perception, with a steady increase in the predictive power from ~−250 ms until stimulus onset, and same topography and optimal phase as the late movement-locked effect (*Figure 4a–b*, bottom). Thus, contrary to the early predictive effect (~−1.4 s), the late movement-locked effect (from ~−0.5 to 0 s) cannot be completely dissociated from the stimulus-locked effect.

The majority of the previous studies showing the influence of pre-stimulus phase on perception have reported the involvement of oscillations around either 7 or 10 Hz (see [*VanRullen, 2016*] for a review and meta-analysis), but not at 4 Hz, as revealed here. To assess whether the present effect is specific for the theta-band, we repeated our analysis for frequencies between 3.5 and 15.5 Hz. *Figure 4c* shows that the results for the movement- (top) and the stimulus-locked (bottom) phases have similar (and selective) spectro-temporal signatures, with theta-band oscillations showing the highest predictive power, and for the most extended epochs. A smaller and non-significant predictive value is found in the alpha-band (see *Figure 4—figure supplement 1* for the complete illustration of the alpha-band effects).

## Discussion

The present work provides two interrelated pieces of evidence pointing to the existence of a movement-locked theta rhythm that affects visual perception. First, theta (~4 Hz) phases relative to movement onset predict perceptual performance. Second, these theta phases are aligned to the onset of the movement. Crucially, these two phenomena unfold during the motor planning stage with an almost identical non-monotonic temporal profile, suggesting their relatedness.

In two distinct epochs of motor preparation, separated by almost 1 s, theta rhythmic activity is phase-locked to the ensuing movement and also perceptually relevant. Intriguingly, this temporal discontinuity seems to reflect two distinct processes that are initiated at different times, but nevertheless share similar spectral features. In fact, theta phases in the early and the late epoch have independent predictive power for perception. This indicates that, despite statistically significant locking to movement onset, the coupling between the early and the late theta phases must be very weak.

Several features distinguish the early effect from the late effect. The most striking one is that, whereas the late effect is proximal to the presentation of the visual stimulus, the early effect foreruns it by more than 1 s. Modulations of behavioral performance by pre-stimulus oscillatory phases have typically been interpreted as reflecting the impact of neuronal excitability at the time of stimulus presentation (*VanRullen, 2016*; *Schroeder and Lakatos, 2009*; *Lakatos et al., 2008*). Indeed, previous studies have shown that oscillatory phase acquires perceptual relevance only a few hundred milliseconds (~100–200 ms) prior to stimulus onset (e.g., [*Busch et al., 2009*; *Hanslmayr et al., 2013*;

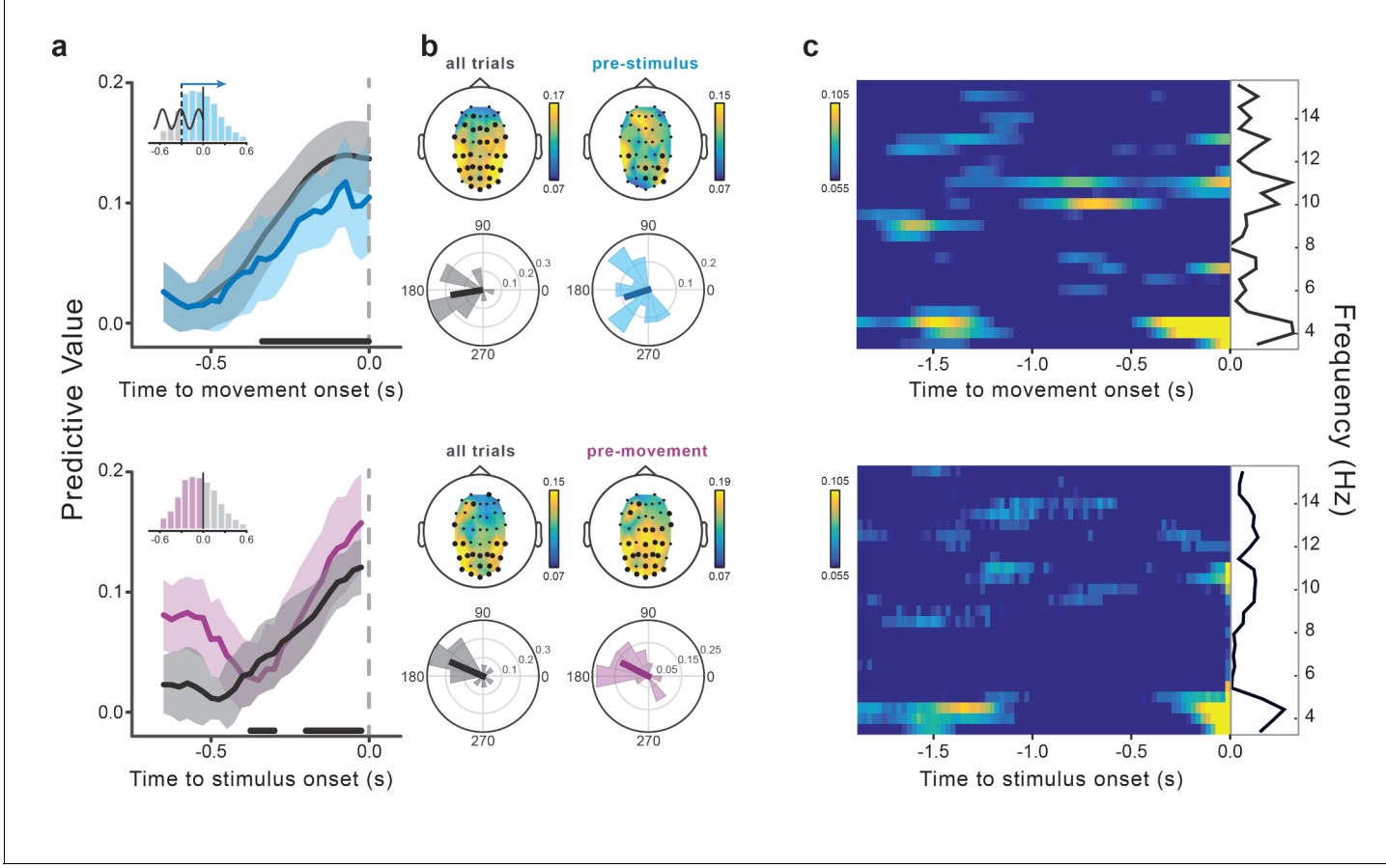

**Figure 4.** Movement-locked and stimulus-locked effects. (a) Time course of the predictive value of theta phases from −0.65 to 0 s relative to movement (top) and stimulus onset (bottom). The dark gray lines represent the predictive value calculated for all trials (i.e., stimuli presented from −0.6 to +0.6 s relative to movement onset; movement-locked data are the same as shown in **Figure 3a**). The light blue line represents the movement-locked effect calculated for trials in which the phase estimation time point preceded the stimulus presentation time (i.e., pre-stimulus or forward extrapolation trials). The pink line represents the stimulus-locked effect calculated only for trials in which the stimuli preceded movement onset. Shaded areas represent the jackknife standard errors. Horizontal bars indicate significant time points for all trials (after FDR correction across space and time points [−0.65–0 s]). The bar histograms in the insets show the distribution of stimulus presentation times relative to movement onset. (b) Topography of the predictive value of theta phase and across-subjects distribution of the optimal phase angle for all trials (dark gray), pre-stimulus trials (light blue; movement-locked effect) and pre-movement trials (pink; stimulus-locked effect). Results are shown at the peak time of the effect (−0.1 and −0.025 s for movement- and stimulus-locked effects, respectively). Significant channels are marked by black bigger circles (FDR-corrected across space and time points [−0.65–0 s] for all trials and across space for the control analyses). (c) Predictive value of oscillatory phase for the perceptual performance as a function of frequency (3.5–15.5 Hz) and time relative to movement (top) and stimulus onset (bottom). The black lines show the predictive value as a function of frequency averaged across the entire time window from −1.9 to 0 s. Time-frequency power plots for both movement- and stimulus-locked data are reported in **Figure 4—figure supplement 2**.

The following figure supplements are available for figure 4:

**Figure supplement 1.** Alpha-band results.

**Figure supplement 2.** Power.

*Busch and VanRullen, 2010*]; for a review see [*VanRullen, 2016*]). Here we report that theta phases during preparation for a movement can predict the perceptual outcome long before stimulus appearance and in a way that is independent from the immediate peri-stimulus phase in the same frequency range (i.e., the late effect). Clearly, these remote oscillatory phases cannot exert any direct influence on sensory processing, at least as would be implied if they reflected different states of neuronal excitability. One possibility is that these early theta phases control a neural switch that brings

some sensory area in a self-sustaining state of higher excitability. This would be consistent with the different topographies of the early and the late predictive effect: the early effect is concentrated over fronto-central sites, whereas the late effect is strongest over parieto-occipital electrodes, which is what would be expected for a neural correlate of visual sensitivity (however, see [*Busch et al., 2009*; *Busch and VanRullen, 2010*; *Dugué et al., 2011*] for similar fronto-central topographies of the ongoing oscillatory activity predicting visual detection). Interestingly, recent studies have described a theta rhythm around 4 Hz in mid- and low-level visual areas (including V1), which is modulated by attention (*Fries et al., 2008*; *Bosman et al., 2012*). Finally, it should be noted that, contrary to the early effect, the late effect is susceptible to multiple potential contaminations, which may result in a complex mixture of signals, and in turn produce an effect topography that reflects different sources.

The current findings are reminiscent of the phase-reset in sensory areas – often accompanied by periodicity in perceptual performance – induced by the presentation of external stimuli (*Romei et al., 2012*; *Mercier et al., 2013*; *Landau and Fries, 2012*; *Fiebelkorn et al., 2013*, *2011*; *Mercier et al., 2015*). However, in our study, no external resetting event can be identified. In fact, both phase-locking and behavioral rhythmicity precede (rather than follow) the event that serves as the temporal reference (i.e., movement onset). Thus, if any discrete resetting event would play a role here, it has to be an endogenous signal (e.g., corollary discharge), operating since the earliest stages of movement planning (>1 s before movement) and carrying an accurate representation of the time of movement initiation. Phase-reset is not, however, the only possible or the most likely mechanism that can explain the present pattern of results. In particular, transient theta oscillatory activity could be a constituent part of movement preparation, reflecting the neuronal processes that lead to motor execution. Alternatively, spontaneous movement initiation (and visual perception) may be influenced by the phase of an ongoing (non-motor) theta rhythm. These two options can hardly be dissociated, as they equally predict the present pattern of results. However, the fact that the movement timing condition (short/long) modulates the temporal dynamics of the predictive effect in a non-trivial way (i.e., not accounted for by the visual cue) suggests that the identified theta rhythm might indeed be involved in the planning of the movement, at least with respect to its timing component.

Previous work has proposed that the motor system can improve the temporal tuning of attentional fluctuations, optimizing information selection when sensory events are rhythmic or, at least, temporally predictable (*Morillon et al., 2015*; *Arnal, 2012*; *Saleh et al., 2010*; *Arnal and Giraud, 2012*; *Morillon et al., 2014*). However, in our study, temporal attention cannot play a role, because the visual stimuli are unpredictable. The present study shows that an endogenous 4 Hz theta rhythm synchronizes perception and action in the absence of any overt rhythmicity either in motor behavior or in external sensory events.

The potential involvement of theta oscillations in sensorimotor processing has been postulated before (*Bland and Oddie, 2001*; *Caplan et al., 2003*) but no compelling neurophysiological evidence had been provided so far. Notably, in all the existing behavioral reports of movement-locked fluctuations in visual performance, the rhythmicity is confined to the theta range ([*Tomassini et al., 2015*; *Benedetto et al., 2016*]; see also [*Wutz et al., 2016*; *Hogendoorn, 2016*] for related findings with eye movements), suggesting that this may indeed be the spectral signature of action-perception coupling.

In conclusion, we have demonstrated the perceptual relevance of movement-locked theta phases as well as their inter-individual consistency. The pattern in the data suggests the existence of two very weakly coupled ~4 Hz oscillations, an early one and a late one, of which the first is likely to be involved in motor timing. These neuronal oscillations in the theta-band may be instrumental in binding action and perception at early stages, possibly providing a common temporal reference frame for integrating sensory information with the emerging motor intention.

## Materials and methods

### Participants

Eighteen healthy participants (11 females; age 24 ± 4.3 year, MEAN±SD), took part in the combined behavioral-EEG experiment. One participant (female) withdrew before completing the experiment

and was excluded from the analysis due to insufficient data. Participants were all naïve with respect to the aims of the study and were all paid (€10/h) for their participation, except for one student who also helped in the data acquisition as part of her Bachelor thesis.

Six healthy participants (2 non-naïve bachelor students; 6 females; age 23 ± 2.1 year, MEAN±SD), took part in the purely behavioral experiment. Participants of both experiments were right-handed (by self-report) and had normal or corrected-to-normal vision. The study and experimental procedures were approved by the local Ethical Review Board (Ethics Committee of the Faculty of Social Sciences, Radboud University, The Netherlands). Participants provided written, informed consent after explanation of the task and experimental procedures, in accordance with the guidelines of the local Ethical Review Board.

No power analysis was used to decide on the sample size, neither for the number of subjects nor for the number of trials per subject. For both experiments, we based our sample size estimations on previous studies showing similar phenomena to the ones under investigation here, i.e., studies relating ongoing EEG phase to perceptual performance (e.g., [*Busch et al., 2009*; *Busch and VanRullen, 2010*]) and one study from our group showing rhythmicity in the time course of behavioral performance aligned to movement onset (*Tomassini et al., 2015*).

## Experimental setup and procedure

Participants sat in a dark room, in front of an LCD monitor (24''; 120 Hz) at a viewing distance of ~57 cm. They held a custom-made isometric joystick in their right hand that allowed to measure hand force along two orthogonal axes via four strain gauges. To avoid excessive fatigue due to the prolonged static posture, participants were instructed to hold the joystick handle with a relaxed grip and to lean their forearm comfortably on the chair arm during the experiment, thus minimizing steady muscle contraction. The joystick was securely fixed to a rigid support to avoid displacement and was positioned a few centimeters below the table so that participants could not see their hand during the experiment.

Participants had to concurrently perform two unrelated tasks: a motor timing task and a visual orientation discrimination task. The motor timing task consisted in pushing the joystick forward with the right hand after one of two specific time intervals (randomly intermingled within each block of trials) initiated by a visual cue. Each trial started with the display of a red fixation cross (size 0.5°) at the center of the screen over a uniformly gray background. After a variable delay (0.8–1.2 s) the fixation cross turned either yellow or green (cue onset) for 0.5 s, then turned red again (cue offset) and stayed red throughout the trial. The color of the fixation cross (yellow/green; counter-balanced across subjects) indicated whether participants had to move their right hand after a short (1.5 s) or a long (2.3 s) time interval. The exact moment when the yellow/green fixation cross turned red again (cue offset) signaled the beginning of the target time interval.

Participants had to judge the orientation of a Gabor patch that could be tilted 45 degrees clockwise or counterclockwise. In this two-alternative forced choice paradigm (2AFC; similar to the task used in [*Tomassini et al., 2015*]), average performance was controlled by presenting the Gabor at a contrast near threshold. The Gabor patch (size 5°, Gaussian envelope SD 0.5°, spatial frequency, 1 c/deg) was briefly displayed (0.016 s; two frames) at the center of the screen. To maximize stimulus sampling during the motor preparatory period, Gabor presentation times were randomly drawn from a uniform distribution ranging from −0.35 to +0.25 s relative to the instructed movement time (i.e. 1.5 and 2.3 s for the *short* and *long* movement timing condition, respectively). At the end of the trial, participants reported verbally whether the Gabor was tilted clockwise or counterclockwise, and did this in response to the question 'What was the orientation?' that appeared on the screen.

Prior to the experiment, participants familiarized themselves with both the visual and the motor task in separate blocks. The trial structure was the same as described above except that participants were required only to judge the Gabor orientation or to execute the movement at the instructed time. The familiarization phase for the visual task (50 trials) provided an indication of the individual contrast threshold (i.e., the contrast yielding ~75% correct responses) that was used to set the initial contrast level for the main experiment. The Gabor contrast was varied on a trial-by-trial basis according to the adaptive QUEST algorithm (*Watson and Pelli, 1983*). Data were fitted with cumulative Gaussian functions and the threshold was derived from the mean of the psychometric function. Due to learning effects, the performance level in the main experiment was continuously monitored and the Gabor contrast was adjusted throughout the experiment to keep performance near threshold.

The percentage of correct responses was calculated after half of the trials in each block (i.e., after 25 trials). The contrast was not changed if the performance was within the desired range, namely between 70% and 80%. The contrast was decreased/increased by 0.6 dB if the performance level was within 80–90% or 60–70%, respectively, or by 1.2 dB if performance was >90% or <60%, respectively.

Participants also had a motor training session in which they learnt to execute brief (~0.5 s) and sharp hand contractions at the instructed moments in time without being told the exact duration of the two target time intervals. The length of the time intervals was illustrated with four example trials (two repetitions for the short and two for the long time interval) in which participants were not required to move but just to pay attention to a brief sound marking the end of the target time interval. After that, participants practiced with the movement (20 trials) and received auditory feedback if they moved too early (≤target interval-0.25 s; high pitch sound) or too late (≥target interval +0.25 s; low pitch sound) with respect to the instructed time, or if they failed to move within ~4 s from the cue offset (intermediate pitch sound). The latter feedback was also given if movement onset could not be detected by the automated algorithm (see below). Auditory feedback was given both during training and during the main experiment. This helped keeping movement timing calibrated throughout the experiment. Twenty practice trials were sufficient for most of the participants to successfully learn the motor task and achieve a stable performance. Three participants required one additional training session (40 trials in total).

A photodiode (2.3 × 2.3 cm) was placed in the top right corner of the monitor and was used to measure the timing of the visual stimulations (cue and Gabor) with millisecond accuracy. A white square (2 × 2 cm) was displayed on the screen at the position of the photodiode (hidden from view) in synchrony with the changes in color of the fixation cross (cue onset/offset) and again with the Gabor onset. Both the signal from the photodiode and that from the isometric joystick were recorded by a National Instruments data acquisition device (sampling rate, 1000 Hz). Cue and Gabor onset times as well as movement onset time were determined on a trial-by-trial basis and these times were used to determine whether auditory feedback had to be given about the participants' motor timing. The onset times for the visual stimuli were derived as the first sample of the photodiode signal exceeding an appropriate threshold. Force onset time was determined as the instant corresponding to the first sample of a series of 15 consecutive samples in which the first derivative of the joystick's voltage signal (along the axis parallel to the direction of the movement) was greater than zero.

The presentation of the stimuli and the data acquisition device were controlled with Psychopy (RRID:SCR_006571).

## Data collection

Data were collected in separate blocks of 50 trials each. For the behavioral experiment, all participants were tested on three separate days (2 hr testing each day) and completed on average 19.6 ± 1.4 (SD) blocks of trials. The task was almost identical to the combined behavioral-EEG experiment (described above) except that participants pushed the joystick in two different directions (right/left) according to a symbolic cue (>,< for right and left, respectively) shown on the screen prior to the beginning of each trial (before the red fixation cross was displayed). The two hand movement directions were randomly intermingled within each block of trials.

For the combined behavioral-EEG experiment, eight participants were tested on two separate days (2 hr testing each day) and completed on average 9 ± 1.3 (SD) blocks of trials, while the remaining nine participants took part in one additional testing day and completed 14 ± 1.6 (SD) blocks of trials.

## Analysis of behavioral data

Data collected in the combined behavioral-EEG experiment (n = 17) and in the behavioral experiment (n = 6) were analyzed separately. Data from the behavioral experiment were pooled across hand movement directions (right/left). To identify the spectral content in the time course of visual performance relative to movement onset, we used logistic regression analysis. This analysis is identical to a generalized linear model (GLM) analysis with a logit link function and a binomial distribution. For each subject's data, we fitted logistic regression models including as predictors a sine and a

cosine of a given frequency in the range from 2.5 to 14.5 Hz (in steps of 0.5 Hz). The probability model behind this analysis can be written as follows:

$$P(Y_i = 1; \text{Time} = t_i) = \text{Logist}[\beta_0 + \beta_1 \sin(2\pi f t_i) + \beta_2 \cos(2\pi f t_i)] \qquad [1]$$

In this equation, $Y_i$ is the response variable for trial i (1 correct, 0 incorrect), $t_i$ is the stimulus presentation time relative to movement onset, $\beta_0$, $\beta_1$ and $\beta_2$ are the fixed-effect logistic regression parameters, and $2\pi f t_i$ is the unwrapped phase of an oscillation with frequency f.

A group-level (random effects) analysis was then performed by testing against zero the average of the participant-specific beta coefficients $\beta_1$ and $\beta_2$ by means of the bivariate Hotelling's T-square statistic:

$$T^2 = n(\bar{\beta}_1, \bar{\beta}_2)' S^{-1} (\bar{\beta}_1, \bar{\beta}_2) \qquad [2]$$

In this equation, $n$ denotes the number of subjects, $(\bar{\beta}_1, \bar{\beta}_2)$ is the sample mean (across subjects) of the vector of logistic regression coefficients $(\beta_1, \beta_2)$, and $S^{-1}$ is the inverse of the sample covariance matrix of these vector-valued regression coefficients. This Hotelling's T-square test provides significant results only if two conditions are fulfilled: (1) the regression coefficients are large (i.e., sufficiently larger or smaller than zero relative to their standard error), indicating consistent association between the phase of the sinusoidal function and perceptual performance, and (2) they have the same sign across subjects (i.e., the phases associated with the highest/lowest performance are aligned across subjects).

The predictive value of the phase was quantified as the norm (Euclidean length) of the sample mean $(\bar{\beta}_1, \bar{\beta}_2)$:

$$\text{Predictive Value} = \sqrt{\bar{\beta}_1^2 + \bar{\beta}_2^2} \qquad [3]$$

The standard error of this predictive value was calculated by means of the Jacknife (**Efron and Gong, 1983**).

The results of the Hotelling's T-square test were not corrected for multiple comparisons across frequencies. However, since movement-locked rhythmicity of visual performance in the theta-band was observed here in two independent samples of participants (n = 6 and n = 17 for the purely behavioral and combined behavioral-EEG experiment, respectively) and also reported in two previous behavioral studies conducted in different laboratories (with different setups and experimental manipulations; see [**Tomassini et al., 2015**] and [**Benedetto et al., 2016**]), we consider it to be a genuine and reproducible phenomenon.

We also performed a fixed-effect statistical test based on a fast Fourier transform (FFT) of the aggregated data from all participants (as in [**Tomassini et al., 2015**]), and we did this separately for the behavioral and the combined dataset (**Figure 2—figure supplement 1**). Trials with stimulus latencies from −0.575 to +0.475 s relative to movement onset were first pooled across all participants and then binned (bin size 0.05 s). We then applied a sliding window stepped by 0.025 s and for each bin we calculated the percentage of correct responses. The resulting behavioral time series was then tapered with a Hanning window and Fourier transformed. We used a permutation technique to evaluate systematic associations between perceptual performance and the stimulus presentation time relative to movement onset. Specifically, we generated a surrogate spectral distribution by permuting the stimulus presentation times 5000 times. Each permutation yielded a surrogate data set that was submitted to the same analysis as performed on the observed data set (binning, Hanning tapering and FFT), producing a distribution of power spectra. The power derived from the FFT output of the observed behavioral time series was then compared at each frequency (from 2.5 to 14.5 Hz in steps of 0.5 Hz) with the reference power distribution. The p-value of the permutation test is yielded by the proportion of values of the reference distribution exceeding the power in the original data set.

For the purpose of illustration, data including stimulus latencies in the range from −0.4 to +0.275 s have been pooled across the six subjects who took part to the behavioral experiment (see **Figure 2a**). On average, the displayed time course in the performance was calculated on 323.6 ± 92 (SD) trials per bin.

## EEG recording and analysis

EEG data were recorded continuously during the experiment (including the motor training phase) with a 64-channel active electrode system (Brain Products GmbH, Gilching, Germany). Electrooculograms (EOGs) were recorded using four electrodes from the cap: FT9 and FT10 were removed from their original scalp sites and placed at the bilateral outer canthi to record horizontal eye movements, and PO9 and PO10 were also removed and placed below and above the right eye to record vertical eye movements. In addition, Fp1 and Fp2 were moved from the cap and used to record electromyographic activity (EMG) from the right arm. Since EMG activity has not been analyzed in this study, no further detail will be provided about the recording procedure.

All electrodes were referenced to the left mastoid. The impedance of the electrodes was kept below 15 kΩ. To accurately synchronize the EEG signal with all the relevant task events (cue, Gabor and hand movement) we used the EEG recording system to acquire also the voltage signal from the photodiode as well as from the isometric joystick. EEG, photodiode and joystick signals were sampled at 1000 Hz.

Analyses of EEG data were performed with the FieldTrip toolbox ([*Oostenveld et al., 2011*]; RRID:SCR_004849). Data were epoched in 4.75 s-long segments roughly corresponding to the entire duration of a trial (from the fixation cross to the question display). The hand force traces for all trials were visually inspected and trials were rejected if movement onset could not be unambiguously detected. EEG data were then manually checked for bad channels and/or artifacts in the time domain after applying a low-pass filter (cutoff frequency, 25 Hz). Bad channels were interpolated with a distance-weighted nearest-neighbor approach. Since channels at the rim of the electrode cap were rather noisy in a large number of participants and for extended periods, we decided to exclude them from the analysis. In total, 45 channels were analyzed, and the following channels were excluded from analysis: AF7, AF8, F7, F8, FT7, FT8, T7, T8, TP7, TP8, P7, P8, and the Right mastoid. Independent Component Analysis (ICA) was used to identify and remove artifacts related to eye movements and heartbeat. Only trials extending at least 2 s before and 0.5 s after movement onset were retained for the analysis. The critical period ranging from −2 to +0.5 s relative to movement onset was further inspected and trials containing residual artifacts in this time window were discarded.

EEG analysis were constrained by the analysis of the behavioral data (see above) as well as by previous evidence showing a predominant theta-band oscillatory component in the time course of perceptual performance aligned to the onset of voluntary movements (*Tomassini et al., 2015*; *Benedetto et al., 2016*). The analysis focused on two main aspects: (1) the influence of the instantaneous EEG theta phase (~4 Hz) time-locked to movement onset on perceptual performance, and (2) phase-locking of theta (~4 Hz) oscillations to movement onset during the motor preparation period. The EEG data were band-pass filtered between 3 and 5 Hz (two-pass Butterworth filter, third-order for each single pass). Thereafter, instantaneous EEG phase time-locked to movement onset was computed by means of the Hilbert transform and the phase values were subsequently down-sampled to 200 Hz. Frequency-resolved analysis (see *Figure 4c*) was performed by applying a sliding window along the frequency axis in the range between 3.5 and 15.5 Hz in steps of 0.5 Hz and with a frequency window length of 2 Hz.

## Quantifying the predictive value of the EEG phase relative to movement onset for perceptual performance

Data were epoched from −1.9 to 0 s relative to movement onset. For each trial $i$ and time point $t$ (in steps of 0.025 s), the phase at stimulus onset, $phase\_at\_stim_i(t)$, was calculated by extrapolation from the EEG-derived phase at time $t$, $phase_i(t)$, which was obtained from the Hilbert transform:

$$phase\_at\_stim_i(t) = phase_i(t) + 2\pi * f * (t\_stim_i - t) \qquad [4]$$

in which $f$ is the center frequency of the band-pass filter (i.e., 4 Hz), and $t\_stim_i$ is the stimulus presentation time (relative to movement onset) in trial $i$.

The phase values at stimulus onset, $phase\_at\_stim_i(t)$, extrapolated from different time points $t$ (every 0.025 s between −1.9 to 0 s), were then used to predict perceptual performance in a similar way as for the analysis of the behavioral data. Specifically, for each subject's data, we fitted logistic regression models including as predictors a sine and a cosine, but now having as an argument

$phase\_at\_stim_i(t)$. The probability model behind this logistic regression analysis can be written as follows:

$$P(Y_i = 1; t) = \text{Logist}[\beta_0 + \beta_1 * \sin(phase\_at\_stim_i(t)) + \beta_2 * \cos(phase\_at\_stim_i(t))] \qquad [5]$$

Note that this regression model was fitted for the different time points $t$.

We then tested at the group level the random effects null hypothesis that the mean across subjects of the logistic regression coefficients ($\beta_1$, $\beta_2$) is equal to zero. For this, we again applied the Hotelling's T-square in the same way as described for the analysis of the behavioral data (see above).

The predictive value of the phases was again quantified as the norm (Euclidean length) of the sample mean ($\bar{\beta}_1$, $\bar{\beta}_2$), see *Equation 3*. The mean and standard error of this predictive value was calculated using the Jackknife (*Efron and Gong, 1983*). The predictive value was estimated for each subsample of participants, omitting each time a different observation $i$, and the standard error was calculated as follows:

$$SE_{jackknife} = \frac{n-1}{n} \sum_{i=1}^{n} \left( predictive\ value_i - \overline{predictive\ value}_{(.)} \right)^2$$

where $predictive\ value_i$ represents the subsample estimate based on leaving out the $ith$ observation, and $\overline{predictive\ value}_{(.)}$ is the average of all subsample estimates.

We corrected for multiple comparisons across both space (EEG channels) and time by controlling the False Discovery Rate (FDR; described in [*Benjamini and Yekutieli, 2001*]). The significant time points shown in *Figures 3a* and *4a* represent those time points for which at least one channel survived the FDR correction.

This analysis was performed for two sets of trials: (1) all trials in which the stimuli were presented from −0.6 to +0.6 s relative to movement onset [96 ± 3% of the total number of trials; MEAN ± SD] and, (2) as a control analysis, the so-called *forward extrapolation trials* (or pre-stimulus trials), which are a subset of trials in (1), namely those in which the visual stimuli were presented *after* the time point at which the phase was estimated. Note that in this analysis, the number of discarded trials rapidly increases as the phase estimation time point is closer to movement onset (where the majority of the stimuli was presented); in the interval between −0.25 and 0 s relative to movement onset, where the effect is strongest, this control analysis is performed with approximately half of the total number of trials (54 ± 2%, MEAN ± SE), and therefore suffers from a reduced statistical power. Statistical evaluation of the predictive value of theta phase in the control analysis was performed for a single time point where the original effect (calculated on all trials) was maximal (i.e., −0.1 s) with FDR-correction for multiple comparisons across space.

The predictive value of the theta phase was also calculated separately for the *short* and the *long* movement timing condition. The difference between the two conditions was statistically evaluated for each time point (relative to movement onset) by means of a group-level permutation test. For each channel and time point we randomly permuted the labels 'short' and 'long' for the individual complex-valued beta coefficients derived from the logistic regression analysis and computed the predictive value as indicated in *Equation 3*. For each permutation, we then calculated the difference between the predictive values for the two conditions, and repeated this 1000 times. P-values were derived as the proportions of permutations yielding a larger difference in the predictive value as compared to the observed difference. We subsequently corrected the p-values for multiple comparisons across space and time such that the FDR was controlled.

We also calculated the predictive value for the *stimulus-locked* theta phases. The only difference with the analysis described above (for the movement-locked phases) is that no phase extrapolation procedure was required. Instead, we used as predictors the instantaneous theta phases estimated in the interval [−0.65–0 s] relative to stimulus onset. Also in this case, the analysis was repeated for all trials and, as a control for possible phase corruption by post-movement activity, only for the pre-movement trials (i.e., the trials in which the visual stimuli preceded the movement). Again, for this control analysis based on pre-movement trials, statistical evaluation was only performed for a single time point where the original effect (calculated on all trials) was maximal (i.e., −0.025 s) with FDR-correction for multiple comparisons across space.

## Quantifying the phase-locking of theta oscillations to movement onset

For every subject, per EEG channel, we quantified the phase-locking to movement onset by means of a measure that is based on the mean across trials of the signal's Hilbert transform, which we denote as *mean resultant vectors* (MRVs; see also *Figure 3f*):

$$MRV_c(t) = \frac{1}{n}\sum_{i=1}^{n} H_{ic}(t) \qquad [6]$$

in which $c$ is the channel index, $t$ denotes time, $i$ is the trial index, $n$ is the number of trials, and $H_{ic}(t)$ is the value of the Hilbert transform for trial $i$, channel $c$, and time $t$ (a complex number). Because all trials were aligned to movement onset (all signals were cut from $-1.9$ to $0$ s relative to movement onset), these MRVs are closely related to the familiar inter-trial phase coherences (ITPC), although they are not normalized for amplitude. Amplitude differences between channels and time points are thus reflected in the MRVs.

For each subject, we also calculated MRVs that were normalized for amplitude. These normalized MRVs were calculated in the same way as the non-normalized ones, but now using $H_{ic}(t)/|H_{ic}(t)|$ instead of $H_{ic}(t)$. The resulting amplitude-normalized MRV (also called inter-trial coherence) was then averaged across subjects. This produced a measure of the inter-individual consistency in the phases locked to movement onset. The topography of this inter-individual phase consistency is shown in *Figure 3e* (top). *Figure 3e* (bottom) shows the topographical distribution of the mean (across subjects) phase angle locked to movement onset.

To statistically evaluate the phase-locking to movement onset we must deal with several factors that have a negative effect on the sensitivity of a statistical test. First, we face the challenge that, in a group analysis (over participants; random effects), the sensitivity of a statistical test is negatively influenced by individual differences in the spatial topography of the phase-locking as well as the preferred phases at which this phase-locking occurs. Second, if we would statistically evaluate the phase-locking across both time and space (EEG channels), this would force us to perform multiple comparison correction over a very large number of statistical tests. We used here a method based on a particular measure of phase reliability at the single-subject level that deals with both these problems (dependence on individual differences in spatial topography and preferred phases, and multiple comparison correction over both space and time). In this method, for every participant, we randomly split the entire set of trials in two partitions of equal size. We then calculate the MRVs for both partitions and, for every participant, we correlate these MRVs across space (EEG channels). The motivation for calculating this spatial correlation is that it is only non-zero if there is systematic locking of the phases to movement onset.

To describe this spatial correlation by means of a formula, we introduce the following vector:

$$\boldsymbol{MRV}^{(p)}(t) = \left[ MRV_1^{(p)}(t), MRV_2^{(p)}(t), \ldots, MRV_C^{(p)}(t) \right] \qquad [7]$$

in which the superscript $^{(p)}$ denotes the partition ($p = 1, 2$), and $C$ denotes the total number of channels. The correlation between the vectors $\boldsymbol{MRV}^{(1)}(t)$ and $\boldsymbol{MRV}^{(2)}(t)$ is calculated as follows:

$$r(t) = \frac{\boldsymbol{MRV}^{(1)}(t) * \boldsymbol{MRV}^{(2)}(t)'}{\sqrt{\boldsymbol{MRV}^{(1)}(t) * \boldsymbol{MRV}^{(1)}(t)'}\sqrt{\boldsymbol{MRV}^{(2)}(t) * \boldsymbol{MRV}^{(2)}(t)'}} \qquad [8]$$

in which $'$ denotes the conjugate transpose. The role of the denominator in *Equation 8* is only to normalize the correlation (i.e., to constrain its amplitude between 0 and 1). It is important to note that the spatial correlation $r(t)$ is complex-valued. However, because the vectors $\boldsymbol{MRV}^{(1)}(t)$ and $\boldsymbol{MRV}^{(2)}(t)$ must have the same phases if there is phase-locking to movement onset, only the real part of $r(t)$ is relevant for the null hypothesis that we want to reject (no phase-locking to movement onset).

The spatial correlation $r(t)$ depends on how the trials are partitioned in two groups. We drastically reduced the dependence on this irrelevant aspect of the calculation by performing 500 random partitions of the trials, and calculating $r(t)$ for each of them. We then calculated the average of

$\text{Real}(r(t))$ over these 500 partitions. This average, which was calculated for each of the participants, served as the input for the statistical test.

To perform a statistical test across participants, we must combine the real parts of the spatial correlations $r(t)$ of all participants. Under the null hypothesis of no phase-locking to movement onset, the expected value of the real part is equal to 0. This null hypothesis can be tested by means of a one-sample t-test, and this is what we did. We corrected for multiple comparisons across the time points in the interval $[-1.9-0 \text{ s}]$ by controlling the FDR.

*Figure 3d* show the time course of the average across subjects of $\text{Real}(r(t))$ and its standard error.

## Availability of data, software, and research materials

Raw data (anonymized), the table of stimulus and response event codes, and the documented analysis scripts are available from the Research Data Repository of the Donders Institute (DI) for Brain, Cognition and Behaviour (http://hdl.handle.net/11633/di.dcc.DSC_2016.00238_334).

## Acknowledgements

The authors declare that they have no competing interests. This work has been supported by the European Research Council (EU-ERC-283567) and the Netherlands Organization for Scientific Research (NWO-VICI: 453-11-001) to PM.

## Additional information

### Funding

| Funder | Grant reference number | Author |
| --- | --- | --- |
| European Research Council | EU-ERC-238-567 | W Pieter Medendorp |
| Nederlandse Organisatie voor Wetenschappelijk Onderzoek | NWO-VICI: 453-11-00 | W Pieter Medendorp |

The funders had no role in study design, data collection and interpretation, or the decision to submit the work for publication.

### Author contributions

AT, Conceptualization, Data curation, Formal analysis, Investigation, Methodology, Writing—original draft, Project administration, Writing—review and editing; LA, Formal analysis, Methodology; WPM, Conceptualization, Supervision, Writing—review and editing; EM, Conceptualization, Formal analysis, Supervision, Methodology, Writing—review and editing

### Author ORCIDs

Alice Tomassini, http://orcid.org/0000-0003-2986-020X
W Pieter Medendorp, http://orcid.org/0000-0001-9615-4220
Eric Maris, http://orcid.org/0000-0001-5166-1800

### Ethics

Human subjects: The study and experimental procedures were approved by the local Ethical Review Board (Ethics Committee of the Faculty of Social Sciences, Radboud University, The Netherlands). Participants provided written, informed consent after explanation of the task and experimental procedures, in accordance with the guidelines of the local Ethical Review Board.

## Additional files

### Major datasets

The following dataset was generated:

| Author(s) | Year | Dataset title | Dataset URL | Database, license, and accessibility information |
|-----------|------|---------------|-------------|--------------------------------------------------|
| Alice Tomassini, Luca Ambrogioni, W Pieter Medendorp, Eric Maris | 2017 | Theta oscillations locked to intended actions rhythmically modulate perception | http://hdl.handle.net/11633/di.dcc.DSC_2016.00238_334 | Available at the Data Repository of the Donders Institute for Brain, Cognition and Behaviour (login required; information on accessing the data can be found at http://www.ru.nl/donders/research/data/user-manual/access-shared-data/) |

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
