## [Decision Letter]

Thank you for submitting your article "Theta oscillations locked to intended actions rhythmically modulate perception" for consideration by *eLife*. Your article has been favorably evaluated by Sabine Kastner (Senior Editor) and three reviewers, one of whom, Benjamin Morillon, is a member of our Board of Reviewing Editors.

The reviewers have discussed the reviews with one another and the Reviewing Editor has drafted this decision to help you prepare a revised submission

Summary:

In this contribution the authors describe a set of behavioral and EEG studies aimed at identifying the neural correlates of a previously described movement-locked theta rhythm which influences the perception of weak visual stimuli. Behavioral results from two experiments confirmed that visual performance locked to the movement onset oscillates at around 4Hz. Then the authors show that EEG theta oscillations are: 1. locked in phase to the onset of the upcoming movement, and 2. locked in phase to the fluctuations of perceptual discriminability. The two theta-phase effects share a common 'non-monotonic' profile, which the authors use as an indication of their relatedness. The authors interpret these findings in terms of an effect of motor planning on sensory perception.

We find the research question addressed by the authors of importance for cognitive neuroscientists interested in the interplay between sensation and action (such as active sensing), but also in the role of brain rhythms in these two cognitive processes. The paradigm, methods and statistical procedures appear sound and appropriate. Despite these strengths of the paradigm and data analysis procedures, we find however that the authors tend to over-interpret their findings, and believe that this study still lacks a set of analysis which truly suggests a "movement-related periodicity of perception in the theta-band". Overall, we believe that this paper would only be novel enough if the authors could convincingly describe the mechanism underlying this 4Hz effect.

Essential revisions:

1) It is unclear whether the observed theta oscillations are action-related in nature. They are 'movement-locked' rather than 'movement-related'. The fact that they are locked to the upcoming action does not mean that they are action-related. The oscillations themselves might constrain movement initiation and perception without being perceptual or motor in nature. For example, this theta rhythm might be central rather than motor, very much like the δ rhythm observed by Wyart and colleagues (2012, Neuron) during cue combination for decision-making. Thus, statements such as "action planning causes […] an endogenous phase modulation of rhythmic sensory activity" or "the same theta rhythm […] synchronizes with the ongoing action plan" (Why is it not the ongoing action that is constrained by (and thus synchronizes with) the ongoing theta rhythm?) are in our opinion not supported by the data.

2) Why did the authors only include results in which the phase information is consistent across participants? As participants have different head shape and EEG cap positions, isn't this choice too restrictive? This choice might for example have led to the failure to replicate the 7 Hz perceptual effects of phase found by Busch and VanRullen (2009, J. Neurosci.), where the authors had been explicit that their analysis did not make this assumption.

3) Concerning the first effect in time (~1.5 s before movement onset):a) Could it be possible to decipher between the two possibilities mentioned by the authors (a single process obscured between -1/-0.4s, or two distinct processes), e.g. by analyzing topographies (lateralization), source-localization (e.g. to show the similar motor origin of both the early and late effects) or power contributions? Alternatively, the authors could investigate the residual fluctuations in the 4 Hz phase at ~1.5 s before movement onset that are uncorrelated with fluctuations in phase around motor onset, to investigate if they still bear predictive power on perception (see Wyart and Tallon-Baudry, 2009, J. Neurosci., for a similar non-monotonic pattern of predictive power tested using such a modulatory procedure).

b) Could it be induced by the cue-evoked activity in the short trials (i.e., due to trial-by-trial variability in overall excitability)? One simple way to refute this hypothesis is to show that this pattern is present even when analyzing only trials with the long interval. Although the authors said that there was no "qualitative difference in the effects for the two sets of trials", it is impossible to make a judgment without seeing the data. Otherwise, the authors could also show that this trial-by-trial variability is somehow correlated with the ongoing EMG signal, which was recorded but not analyzed.

c) The relatedness between the two effects of theta phase on perception and movement onset is indeed striking, but might undermine the genuineness of the early (~1.5 s pre-movement) effect for perception. The authors actually mention this point in the Discussion, but in the opposite direction: "the predictive value of the late phases could hence be inherited by the early phases, just by virtue of their systematic relation". But couldn't it be the opposite? There is no reason to assume that, simply because the early phases precede the late ones, they bear an a priori more "direct" effect on perception.

4) What is the difference between Figure 3 and Figure 4? If we understood the methods correctly, the dark line should represent the same information in both figures. If this is so, why is the topography different between Figure 3 and Figure 4 (and what does this difference could mean)? Related to this, why the investigation of multiple frequencies performed in Figure 4 was not done for results of Figure 3? An analysis of phase-locking (MRV), in particular to the α frequency would be valuable. Indeed, the authors focus on theta phase on the basis of behavioral results and previous contributions. However, both their behavioral and EEG results point also toward a contribution of α frequency (Figure 2 and Figure 4 – the effect in Figure 4 being as strong at 4 and 11 Hz, and the 11 Hz peak being not present in the stimulus locked control analysis).

---

## [Author Response]

*Essential revisions:*

*1) It is unclear whether the observed theta oscillations are action-related in nature. They are 'movement-locked' rather than 'movement-related'. The fact that they are locked to the upcoming action does not mean that they are action-related. The oscillations themselves might constrain movement initiation and perception without being perceptual or motor in nature. For example, this theta rhythm might be central rather than motor, very much like the δ rhythm observed by Wyart and colleagues (2012, Neuron) during cue combination for decision-making. Thus, statements such as "action planning causes […] an endogenous phase modulation of rhythmic sensory activity" or "the same theta rhythm […] synchronizes with the ongoing action plan" (Why is it not the ongoing action that is constrained by (and thus synchronizes with) the ongoing theta rhythm?) are in our opinion not supported by the data.*

We thank the reviewers for having raised this important point. We agree that our results do not allow us to conclude that motor processes cause a modulation of the phase of perceptually relevant oscillations. Our results demonstrate a coupling between motor *output* and the phase of 4 Hz rhythmic activity which also affects perception, but the exact neurophysiological origin or nature of this 4 Hz rhythm is not known.

This issue has been extensively discussed in a previous behavioral paper with the same first author (see the discussion in Tomassini et al., 2015), and we now took the same care in considering multiple possible neurocognitive mechanisms underlying our findings. We have therefore removed from the text any misleading reference to a causal role of the motor system. However, some of the additional analyses proposed by the reviewers (see point 3 for details) have yielded results which, although not conclusive, provide some important clues in support of an oscillatory phenomenon that is related to movement planning. The new results are shown in figure supplements and are extensively described and discussed in the main text.

The paragraph below (extracted from the Discussion section) represents a core passage that conveys the main conclusion of our study:

“[…]Phase-reset is not, however, the only possible or the most likely mechanism that can explain the present pattern of results. […] However, the fact that the movement timing condition (short/long) modulates the temporal dynamics of the predictive effect in a non-trivial way (i.e., not accounted for by the visual cue) suggests that the identified theta rhythm might indeed be involved in the planning of the movement, at least with respect to its timing component.”

*2) Why did the authors only include results in which the phase information is consistent across participants? As participants have different head shape and EEG cap positions, isn't this choice too restrictive? This choice might for example have led to the failure to replicate the 7 Hz perceptual effects of phase found by Busch and VanRullen (2009, J. Neurosci.), where the authors had been explicit that their analysis did not make this assumption.*

This remark raises two points, one pertaining to our statistical test, and one pertaining to our inability to replicate a previously reported effect. Below, we will address these two points separately.

A) A random-effect test of a null hypothesis pertaining to a complex-valued association measure

Compared to other methods for statistically evaluating the association between EEG/MEG-measured oscillatory phase and behavioral performance, our method tests a different null hypothesis, which can be considered as more restrictive. Specifically, we test the null hypothesis that the population average (across participants) of our complex-valued association measure (the pair of logistic regression coefficients represented as a point in the complex plane) equals zero. This is also called a random-effects statistical test, and is to be distinguished from the fixed-effects statistical tests that are more common in this field (e.g., see VanRullen, 2016). A fixed-effects statistical test pertains to the *sample* average of some association measure, and the population to which one generalizes is the population of all *trials* that could be drawn from this sample of participants. The latter is reflected in the fact that the reference distribution is obtained by permuting trial-specific phases relative to the trial-specific behavioral responses.

Because we test a null hypothesis that pertains to the population average of a number in the complex plane (reflecting both amplitude and phase), we implicitly test the consistency across participants of these numbers’ phases. This differs from the more common fixed-effects statistical tests that involve reference distributions for the sample average of participant-specific association measures that only reflect the strength of the association and not its preferred phase. For this reason, our statistical analysis can be considered as more restrictive, and this qualifies the statistically significant result that we obtained: we reject a null hypothesis that pertains to phase consistency within a population of participants.

However, our random-effects statistical analysis cannot easily be adapted to test a null hypothesis that only pertains to the population average of the *amplitude* of our complex-valued association measure. The reason for this is that, with a finite number of trials, the amplitude of this association measure is always positively biased.

B) Replication of a previously reported association between perception and phase of a 7 Hz ongoing rhythm

Following the reviewers’ suggestion, we have reanalyzed our data with the approach used in Busch & VanRullen (2009). This approach involves a statistical test of a measure that reflects the strength of the association (and not its preferred phase). Specifically, it reflects the degree of phase opposition between the trials corresponding to the two different perceptual outcomes (in our case, correct/incorrect orientation discrimination). In Busch and VanRullen (2009), phase opposition was quantified by the *phase bifurcation index* (PBI), which has recently been shown (by the same research group) to have inferior statistical power as compared to an alternative quantification, the *phase opposition sum* (POS; see VanRullen, 2016). Therefore, we used the POS instead of the PBI, but for the remaining aspects, we followed exactly the same statistical procedure as in Busch and VanRullen (2009). Unfortunately, we did not find a consistent pattern of results, and this was reflected by the fact that, following FDR correction, no time-frequency point was significant.

Two reasons might explain the failure to replicate the previous findings by VanRullen and other groups. In fact, our dataset has two important differences as compared to the majority of the previous studies:

Because of the use of a 2AFC design, in our study, the number of trials for the two perceptual categories is highly unbalanced (by design, 75% for correct and 25% for incorrect orientation discrimination). This contrasts with the YES/NO detection paradigm, which typically yields an equal number of hits and misses when stimuli are presented at threshold level.

The phase is not uniformly distributed across the entire set of trials, as a result of the phase-locking to the onset of the movement (especially in the theta-band) combined with the fact that the visual stimuli were concentrated around movement onset (see Figure 3 for the distribution of stimulus presentation times relative to movement onset).

The unbalanced number of trials has been shown to be a factor that reduces the statistical power of the phase opposition analysis (see VanRullen, Front Neurosci. 2016 for a detailed discussion). In addition, the phase opposition measure may very well be suited for studying the perceptual relevance of ongoing oscillatory activity (because of the uniformity of the ongoing phases), but not phases that are locked to an event. Thus, the unbalanced number of trials and the non-uniformity of the phases in our dataset may make the phase opposition analysis underpowered, and therefore not suited to identify the relevance of the oscillatory phase for perceptual performance.

*3) Concerning the first effect in time (~1.5 s before movement onset):a) Could it be possible to decipher between the two possibilities mentioned by the authors (a single process obscured between -1/-0.4s, or two distinct processes), e.g. by analyzing topographies (lateralization), source-localization (e.g. to show the similar motor origin of both the early and late effects) or power contributions? Alternatively, the authors could investigate the residual fluctuations in the 4 Hz phase at ~1.5 s before movement onset that are uncorrelated with fluctuations in phase around motor onset, to investigate if they still bear predictive power on perception (see Wyart and Tallon-Baudry, 2009, J. Neurosci., for a similar non-monotonic pattern of predictive power tested using such a modulatory procedure).*

We thank the reviewers for their suggestions. In fact, the additional analysis proposed by the reviewers has proven valuable in better characterizing the relationship between the early and the late effect. The new evidence has revealed that some of our original speculations were incorrect.

Indeed, the early and the late effect have different topographies, respectively, fronto-central and occipito-parietal (see Figure 3 and Figure 4). Unfortunately, the mere topographic difference does not provide conclusive evidence in favor of two distinct neural sources. For instance, one would also observe different topographies in case of a travelling wave, possibly generated by a deep source that changes orientation over time. Because the electromagnetic inverse problem is inherently ill-posed, we cannot expect that source reconstruction will provide a definite answer to the question of whether the early and the late effect are generated by distinct neural sources.

To investigate this issue in a different way, we have followed the suggestion of the reviewers and tested whether the early and the late theta phases independently predict perception. The results clearly indicate that this is the case, and this points to two de-coupled oscillatory phenomena that both are relevant for perception.

Specifically, we have run logistic regression analyses in which we use both the early (estimated at -1.4 s) and the late theta phase (estimated at -0.1 s) as predictors of the perceptual performance. We performed this analysis in two different ways. In the first way, the logistic regression analysis was performed channel-by-channel (as described in the paper for the original analysis), and we used the early and the late theta phases estimated at corresponding EEG channels. Both the early and the late theta phases predict perception, with an almost identical pattern as the original analysis, both with respect to topography and effect size. This indicates that the two effects are independent.

However, because the early and the late effect have different topographies, the channel-by-channel analysis described in the previous paragraph is unlikely to include the relevant phases (the ones for which one has to partial out) for every channel. Therefore, we also ran a second analysis, in which we partialled out for the phases of the other effect measured at its best channel: the channel with the highest effect size in the original analysis for the early (AF3) and the late (CP5) effect. Also in this case, both effects are significant (early effect, AF3: p<0.001; late effect, CP5: p<0.001), confirming their independence.

The results of these analyses are now described in the text (reported below) and illustrated in Figure 3—figure supplement 1.

“The temporal discontinuity in the theta phases’ predictive value and their alignment to movement onset raises a fundamental question about the nature of the underlying oscillatory phenomenon. […] This is clear evidence for two de-coupled oscillatory phenomena in the theta range.”

*b) Could it be induced by the cue-evoked activity in the short trials (i.e., due to trial-by-trial variability in overall excitability)? One simple way to refute this hypothesis is to show that this pattern is present even when analyzing only trials with the long interval. Although the authors said that there was no "qualitative difference in the effects for the two sets of trials", it is impossible to make a judgment without seeing the data. Otherwise, the authors could also show that this trial-by-trial variability is somehow correlated with the ongoing EMG signal, which was recorded but not analyzed.*

Done. In Figure 3—figure supplement 2, we now show the predictive value of the theta phase calculated separately for the *long* and *short* trials. The pattern of results clearly rules out any potential confound of the visual cue, and, importantly, provides a clue that the theta oscillatory activity that we have identified as perceptually-relevant is also related to the temporal aspects of the planning of the movement.

Below we report a paragraph extracted from the Results section describing the main findings and its implications:

“Because of the motor timing component of our dual task, the onset of the movement follows the visual cue by a certain amount of time, which, for the short condition, almost coincides with the period at which the early theta effect is observed (cue-movement interval: 1.5 ± 0.2 s; MEAN ± SD). […] Thus, the modulation of the predictive effect by the target movement time (long versus short) suggests that the perceptually relevant theta phases are generated by a process that is involved in movement timing.”

*c) The relatedness between the two effects of theta phase on perception and movement onset is indeed striking, but might undermine the genuineness of the early (~1.5 s pre-movement) effect for perception. The authors actually mention this point in the Discussion, but in the opposite direction: "the predictive value of the late phases could hence be inherited by the early phases, just by virtue of their systematic relation". But couldn't it be the opposite? There is no reason to assume that, simply because the early phases precede the late ones, they bear an a priori more "direct" effect on perception.*

As explained in detail in the reply to point 3, the results yielded by the new analysis, suggested by the reviewers, clarifies this aspect. The early effect is indeed genuine, in the sense that it is not a consequence of phase coupling between the early and late theta phases. This result, together with the difference in topography between the early and late effect, allowed us to decipher between the two formerly proposed interpretations (i.e., a single process obscured between -1/-0.4s, or two distinct processes), strongly pointing to two distinct and de-coupled oscillatory phenomena. The genuine nature of the early effect and its modulation by the movement timing condition (see reply to point 3 and Figure 3—figure supplement 1 and Figure 3—figure supplement 2) further suggests the possible involvement of the early theta activity in movement planning. We have now modified the text thoroughly in light of this new important piece of evidence.

*4) What is the difference between Figure 3 and Figure 4? If we understood the methods correctly, the dark line should represent the same information in both figures. If this is so, why is the topography different between Figure 3 and Figure 4 (and what does this difference could mean)? Related to this, why the investigation of multiple frequencies performed in Figure 4 was not done for results of Figure 3? An analysis of phase-locking (MRV), in particular to the α frequency would be valuable. Indeed, the authors focus on theta phase on the basis of behavioral results and previous contributions. However, both their behavioral and EEG results point also toward a contribution of α frequency (Figure 2 and Figure 4 – the effect in Figure 4 being as strong at 4 and 11 Hz, and the 11 Hz peak being not present in the stimulus locked control analysis).*

The dark gray lines in Figure 3 and Figure 4 (upper panel) show the same data, except that Figure 3 illustrates the entire time course (from -1.9 to 0 s), whereas Figure 4 shows only the last part close to movement onset (i.e., the late effect from -0.65 to 0 s). The topographies reported in Figure 3 and Figure 4 are calculated at two different time points: Figure 3 shows the topography of the early effect calculated at -1.4 s, while Figure 4 shows the topography of the late effect calculated at -0.1 s.

We agree with the reviewers that the difference in topography between the early and the late effects is an interesting finding that was insufficiently stressed in the original version of the manuscript. As discussed extensively in the reply to point 3, we have now performed an additional analysis which suggests that the early and late effects reflect two distinct processes that not only have different topographies but also independent predictive power for perception. The topographies corresponding to the early and late (independent) predictive values of the theta phases are now shown in a new figure (Figure 3—figure supplement 1).

As regards the spectral specificity issue and, in particular, the possible relevance of α-band oscillations:

In the behavioral data, we found an α-band component in the so-called behavioral dataset (n=6, approx. 20 blocks of trials per participant). However, the α-band component was absent in the combined behavioral-EEG dataset (n=17, approximately 11 blocks of trials per participant). Because of the absence of an α-band effect in the latter dataset and of the previously reported selective theta-band periodicity in the perceptual performance (Tomassini et al., 2015; Benedetto et al., 2016), we focused our EEG analysis specifically on the theta-band.

At the physiological level, the α-band effect reported in Figure 4, although it is of comparable strength as the 4 Hz effect over the entire time window from -1.9 s to 0 s (rightward inset in Figure 4), does not reach statistical significance at any point in time. Apparently, the individual differences in the α-band effect are substantially larger than those in the theta-band effect.

As suggested by the reviewers, we have now performed the same analysis based on split-half phase reliability to ascertain whether α oscillations are locked to movement onset. These results are shown in a new figure (supplement figure to the main Figure 4) together with the results for the predictive value of α phase for perception. The α-band phases are locked to movement onset, but only in a brief period just before movement execution (from -0.15 to 0 s).

Overall, and in line also with the existing behavioral evidence (Tomassini et al., 2015; Benedetto et al., 2016), we can conclude that the movement-locked periodicity in visual performance is predominantly in the theta band and there is only weak evidence in favor of an α-band rhythm in the relevant data of the present study.